# Early Season Drought Largely Reduces Grain Yield in Wheat Cultivars with Smaller Root Systems

**DOI:** 10.3390/plants8090305

**Published:** 2019-08-27

**Authors:** Victoria Figueroa-Bustos, Jairo A. Palta, Yinglong Chen, Kadambot H.M. Siddique

**Affiliations:** 1The UWA Institute of Agriculture, and UWA School of Agriculture and Environment, The University of Western Australia, LB 5005, Perth, WA 6001, Australia; 2CSIRO Agriculture & Food, Private Bag No. 5, Wembley, WA 6913, Australia

**Keywords:** early season drought, root system size, phenology

## Abstract

In the Australian grainbelt, early winter rainfall has declined during the last 30 years, and farmers sow their crops dry, increasing the risk of early season drought. This study aimed to examine whether differences in the root systems were associated with tolerance to early season drought. Three wheat cultivars with different root systems were grown in 1 m columns in a glasshouse. Immediately after sowing in dry soil, 440 mL water (equivalent to 25 mm rainfall) was supplied to each column, and no water was added to induce the early-season drought for the next 30 days. Shoot and root traits were measured at the end of the early season drought, anthesis and at maturity, respectively. The restricted water supply reduced Ψ_leaf_, stomatal conductance, leaf photosynthetic rate, shoot and root biomass. Early season drought delayed phenology in all cultivars, but there was recovery of root and shoot biomass at anthesis in all three cultivars. Leaf area and shoot biomass at anthesis in Bahatans-87 (large root system) recovered better than Tincurrin (small root system). At maturity, early season drought reduced grain yield more in Tincurrin than Bahatans-87. The slow phenology of Bahatans-87 allowed greater recovery after the drought in leaf area and shoot biomass, which may explain the smaller reduction in grain yield after early season drought.

## 1. Introduction

Climate change is altering the pattern of the rainfall in the Australian grainbelt, affecting wheat production [1]. In the Australian grainbelt, drought spells often occur early in the season [2], mid-season (from stem elongation to two weeks before flowering) [3] and from flowering (terminal drought) [4]. Early sowing is common practice in the Australian grain belt, allowing wheat crops to escape the severe effects of terminal drought on grain yield [5]. However, early winter rainfall in the Mediterranean-type climate of Australia has declined and in response, 82% of wheat growers are now sowing all their crops into dry land [2]. Wheat crops sown into dry soil will germinate at the first rainfall, which could leave crops vulnerable to 20–32 days of drought after emergence (i.e., early season drought, ESD). While seedlings subjected to ESD often survive [6], their growth and development are significantly slowed, which delays anthesis and reduces biomass and grain yield [7]. ESD reduces the rate of leaf net photosynthesis, tiller number, leaf area, aboveground biomass and grain yield [8,9,10] as well as below ground traits, such as, total root length and root surface area, increases the root: shoot ratio [11].

The size of the root system is determined by total root biomass and cumulative root length [12,13] is considered together with leaf area, tiller survival and time to anthesis, as candidate traits that should be examined under early season drought [7]. Under field conditions, larger root systems increased wheat grain yield [14,15] and root length has been positively correlated with yield under water stress [16]. However, selection for increasing grain yield in Australian wheat varieties has unintentionally selected for less total root length, root length density and root biomass; the reduced root system size is more efficient at nitrogen (N) uptake [17,18].

This study examined whether differences in root systems are associated with differences in tolerance to ESD. It was hypothesised that wheat cultivars with large root systems are less affected by ESD with better growth and biomass recovery by anthesis than wheat cultivars small root systems. To test this hypothesis, three wheat cultivars with different-sized root systems were exposed to 30 days of ESD. Differences in shoot and root traits were analysed at the end of the drought period; after recovery at anthesis and at final harvest.

## 2. Materials and Methods 

### 2.1. Plant Materials and Growing Conditions

Three wheat cultivars with different sized root systems, identified from a previous rhizobox study, were used in this study: Bahatans-87 (large root system), Harper (medium root system) and Tincurrin (small root system). The selection of these three cultivars was based on two criteria: total root length and total root biomass at 63 days after sowing (DAS) when Tincurrin was in booting stage meanwhile Bahatans-87 and Harper were in stem elongation. The faster phenological development of Tincurrin allow this cultivar reached the maximum root biomass earlier than Bahatans-87 and Harper.

The three cultivars were grown in polyvinyl chloride columns (0.15 m diameter, 1.0 m deep) with a long sleeve clear plastic bag (150 µm thick with 24 small holes in the bottom to facilitate drainage) inserted into each column for the ease of root recovery at harvest. Each column was filled with 26 kg of soil at a bulk density of 1.47 g cm^–3^ over a 5 cm layer of gravel at the bottom to facilitate drainage. The soil was a mixture (75:25) of a reddish-brown sandy clay loam (Red Calcic Dermosal) [19], obtained from the top soil (0–15 cm) of a field site at Cunderdin, Western Australia, and washed and air-dried river sand to improve drainage [17,20]. Fertiliser equivalent to 60 kg ha^–1^ N, 77 kg ha^–1^ P, 71 kg ha^–1^ K and trace amounts of micronutrients (S, Cu, Zn, Mo and Mn) was mixed homogeneously into the top 0.1 m soil layer in each column before sowing. Fertilizer dose at sowing corresponds to the optimal nutrient supply for wheat crops in the Cunderdin district in Western Australia, from where the soil was collected for the experiment [21]. Daily inspection of wheat plants did not show any visual nutrient deficiency symptoms. Five seeds were sown per column, and thinned to two plants per pot at the 1–2 leaf stages. A water soluble fertiliser with Mg, Cu, Zn, Mo, S and other micronutrients was supplied trough irrigation at 44 and 61 DAS as required.

The experiments were conducted in an evaporative cooled glasshouse at The University of Western Australia, Perth, Australia (31°93′S, 115°83′E) from May to November 2018, with an average air temperature of 17 °C (ranged from 7–26 °C), relative humidity of 63% and natural day light (photoperiod) of 11 to 12 h with an average maximum photosynthetic photon flux density of 935 ± 19 µmol m^₋2^ s^₋1^ measured at the plant level at 13:00 h. 

### 2.2. Experimental Design and Treatments

The three cultivars were grown in a row column design with two factors (cultivar × water) and five replicates. Ninety columns filled with dry soil were divided in two groups of 45 (15 columns per cultivar). In the first group (well-watered, WW), the soil in each column was slowly watered by hand to saturation and left to drain for 48 h to obtain the column soil water capacity. For both groups, fertiliser was applied and homogenously mixed in the top 0.1 m of the soil of each column before sowing; seeds were sown into the dry soil in the second group (ESD). Immediately after sowing, both groups received simulated rainfall of about 25 mm to induce germination (Figure 1). 

The soil of the well-watered treatment was slow watered by hand twice a week to keep the soil close to field capacity and avoid excessive drainage. The ESD treatment received no additional water 30DAS to induce a water deficit. During this period, the volumetric water content was measured twice a week with a 15 cm soil moisture probe (Hydrosense, Campbell, Australia) (Figure 2a,b). The ESD treatment was terminated at 30 DAS, when the plants showed permanent wilting symptoms, by slowly hand watering the soil in the columns to field capacity. All the plants were then watered by hand on alternate days close to column holding capacity until physiological maturity-Z91 [22].

### 2.3. Measurements

Plant phenology was monitored regularly using the scale of Zadoks and Chang [22]. Days to booting (Z49), and physiological maturity (Z91) were recorded.

At 30 DAS plants in the ESD treatment showed permanent wilting symptoms, measurements of leaf water potential (Ψ_leaf_), stomatal conductance (g_s_), transpiration and leaf net photosynthesis rate were taken on the top fully expanded leaf of the main stem on five replicate plants between 10:30 am and 1:30 pm on days with clear sky. Leaf net photosynthesis, g_s_ and transpiration were measured with a LI-COR gas-exchanged system (LI-6400, LI-COR Bioscience, Lincoln, NE, USA) with LED light source on the leaf chamber. In the LI-COR cuvette, CO_2_ concentration was set to 380 µmol mol ^–1^ and LED light intensity 900 µmol m ^–2^ s ^–1^, which is the average saturation intensity for photosynthesis in wheat [23]. Immediately after these measurements, Ψ _leaf_ was measured using a Scholander pressure chamber (model 1000, PMS Instrument Co., Albany, OR, USA). The flag leaf was loosely covered with a plastic sheath before excision and during the measurement to avoid evaporation [24].

Aboveground measurements—leaf area (LA), leaf biomass, specific leaf area (leaf area per unit leaf weight, SLA), tiller number and shoot biomass—were made on 30 DAS (just before the early season drought treatment was watered), 50% anthesis, and maturity. At each sampling time, the shoots were cut from the roots at the crown, the number of tillers recorded and stems and leaves separated. Leaf area was measured using a portable leaf area meter (LI-3000, Li-COR Bioscience, Lincoln, NE, USA). Stems and leaves were dried separately in an oven at 70 °C for 48 h and then weighed. 

Immediately after harvesting the shoots, the plastic bag in each column was pulled away and opened. The soil profile was sampled in 0.2 m sections from the top by cutting the soil with a carbon steel blade. The roots in each 0.2 m section were recovered from the soil by washing through a 1.4 mm sieve to produce a clean sample [25]. The recovered roots from each 0.2 m soil section were placed in plastic bags at 4 °C until being scanned at 400 dpi per mm (Epson Perfection V800, Long Beach, CA, USA) to quantify root morphological traits (the first two harvests only), including root length, root length density (root length per unit of soil volume; RLD) and specific root length (root length per unit of biomass; SRL). The root samples were dried after scanning as per the shoot samples, to obtain root biomass. Root images were analysed using WinRHIZO Pro Software (v2009, Regent Instrument, Quebec, QC, Canada) [26]. Total RLD was calculated as the total root length divided by the soil volume. The distribution of RLD in the soil profile was calculated as the root length in 0.2 m sections from the top to the bottom of each column divided by the soil volume of the corresponding section (0.0035 m^3^). Specific root length, an indirect measure of the thickness of the root system, was estimated as total root length divided by total root biomass [17,27].

At the final harvest, the number of spikes per plant was counted. Spikes were separated from shoots, oven dried at 60 °C for 48 h before being hand-threshed. The number and weight of grains per plant were recorded. Harvest index (HI) was calculated as the ratio of grain yield to shoot biomass.

### 2.4. Statistical Analysis 

The data from each harvest were analysed using two-way ANOVA in the statistical software GenStat 18th edition (VSI International, Hemel Hempstead, UK, 2015), followed by the least significant difference (l.s.d.) test for multiple comparisons.

## 3. Results

### 3.1. Soil Water Content, Water Potential, Stomatal Conductance and Leaf Photosynthesis

In the first 30 days after sowing (DAS), the volumetric soil water content in the top 0.15 m of the soil profile in the well-watered treatment remained high; range 28–34% for the three cultivars (Figure 2a). Leaf water potential (Ψleaf) in the well-watered plants was maintained at about −0.46 MPa for the three cultivars (Figure 2c). In the ESD treatments, volumetric soil water content in the top 0.15 m of the soil profile decreased continuously from about 19% to 10% in the first 30 DAS (Figure 2b), and Ψleaf decreased to –2.46 MPa in Bahatans-87 and about –1.94 MPa in Harper and Tincurrin (Figure 2c). In the first 30 DAS, ESD reduced Ψleaf by a factor of 5.6 in Bahatans-87 and 4.2 in Harper and Tincurrin (*p* < 0.001). In the well-watered treatment, the three cultivars had a similar gs of ~816 mmol m ^–2^ s ^–1^, but g_s_ declined in the ESD treatment ~92 mmol m ^–2^ s ^–1^ (~90% reduction) (*p* < 0.001; Figure 2d). When soil water was sufficient, the leaf photosynthetic rate was ~25 µmol m^–2^s^–1^ in all cultivars (Figure 2e), but declined under ESD to 9.15 µmol m^–2^ s^–1^ (65% reduction) in Harper and ~4.3 µmol m^–2^ s^–1^ (~83% reduction) in Bahatans-87 and Tincurrin (*p* < 0.001; Figure 2e).

### 3.2. Phenology 

In the well-watered treatment, time to booting (Z49) was 19 and 26 days earlier in Tincurrin than Bahatans-87 and Harper, respectively (Table 1). The ESD treatment delayed the time to booting by 7 days in Bahatans-87 and Tincurrin, and 10 days in Harper (*p* < 0.01). Differences in time to anthesis (Z61) between cultivars and treatments followed the same trend as time to booting (Table 1). ESD delayed anthesis by 13 days in Harper and 7 days in Bahatans-87 and Tincurrin (*p* < 0.01). Regardless of the watering treatment, Tincurrin reached physiological maturity (Z91) three days earlier than Bahatans-87 and 25 days earlier than Harper. ESD delayed physiological maturity by 4 and 10 days in Harper and Tincurrin, respectively (*p* < 0.01). Independent of the watering treatment, grain filling in Bahatans-87 was 25 and 13 days shorter than Harper and Tincurrin, respectively (Table 1).

### 3.3. Shoot Traits

At 30 DAS well-watered Bahatans-87 had 25% more leaf area than Harper and Tincurrin (Table 2). In the first 30 DAS, ESD reduced leaf area five-fold in Bahatans-87 and three-fold in Harper and Tincurrin (*p* < 0.01; Table 2). At anthesis (Z61), well-watered Harper had 25% and 41% more leaf area than well-watered Bahatans-87 and Tincurrin, respectively; in the ESD treatment, Harper had 33% and 51% more leaf area than Bahatans-87 and Tincurrin, respectively (Table 2). By anthesis, ESD had reduced leaf area by 15% in Bahatans-87 and 23% in Tincurrin (*p* < 0.05), but had no effect on leaf area in Harper (Table 2).

At 30 DAS well-watered Tincurrin plants had 12% more SLA than Bahatans-87 and Harper (Table 2); in the ESD treatment, Tincurrin and Harper had 25% more SLA than Bahatans-87. At 30 DAS, ESD reduced SLA by 41%, 22% and 29% in Bahatans-87, Harper and Tincurrin, respectively (*p* < 0.05). At anthesis, well-watered Tincurrin had 25% and 20% more SLA than Bahatans–87 and Harper, respectively (Table 2); in the ESD treatment Tincurrin had 25% and 22% more SLA than Bahatans-87 and Harper, respectively. ESD had no effect on SLA at anthesis in any cultivar. 

At 30 DAS, well-watered Bahatans-87 had one and two more tillers per plant than Harper and Tincurrin, respectively (*p* < 0.001), while none of the cultivars had produced tillers in the ESD treatment (Table 2). At anthesis, well-watered Bahatans-87 and Harper had nine and ten more tillers per plant than Tincurrin (Table 2); in the ESD treatment Bahatans-87 and Harper had seven and nine more tillers per plant than Tincurrin. ESD reduced the number of tillers at anthesis by 18%, 11% and 13% in Bahatans-87, Harper and Tincurrin, respectively (*p* < 0.01; Table 2). 

At 30 DAS, well-watered Bahatans-87 had 25% and 38% more shoot biomass than Harper and Tincurrin, respectively (Figure 3a), while all cultivars had similar low shoot biomass in the ESD treatment. In the first 30 DAS, ESD reduced shoot biomass by 70% in Bahatans-87, 60% in Harper and 55% in Tincurrin (*p* < 0.01; Figure 3a). At anthesis, well-watered, Bahatans-87 had 8% and 45% more shoot biomass than Harper and Tincurrin, respectively (*p* < 0.01; Figure 3b), and 7.2% and 61.6% more, respectively, in the ESD treatment (*p* < 0.01). Shoot biomass at anthesis did not differ between the well-watered and ESD treatments in Bahatans-87 and Harper, but Tincurrin had 31% less shoot biomass in the ESD treatment (Figure 3b; *p* < 0.01). 

### 3.4. Root Traits

At 30 DAS, well-watered Bahatans-87 and Tincurrin had similar total root length per plant, while Harper had 7 m more than the other two cultivars (*p* < 0.01). In ESD treatment at 30 DAS, the three cultivars had similar low total root lengths (Figure 4a). In the first 30 DAS, ESD reduced total root length by 75% in Bahatans-87, 54% in Harper and 70% in Tincurrin (*p* < 0.01; Figure 4a). At anthesis, well-watered Bahatans-87 had 17% and 42% more total root length than Harper and Tincurrin, respectively (*p* < 0.001); in the ESD treatment Bahatans-87 had 9% and 46% more total root length than Harper and Tincurrin, respectively. ESD reduced total length at anthesis by 33% in Bahatans-87, 27% in Harper and 38% in Tincurrin *(p* < 0.001; Figure 4b).

At 30 DAS, well-watered Bahatans-87 and Tincurrin had similar root biomass per plant, but Harper had 17% more (*p* < 0.05). In the ESD treatment at 30 DAS, the three cultivars had similar total root biomass (Figure 4c). In the first 30 DAS, ESD reduced total root biomass in all three cultivars by about 30% (*p* < 0.01; Figure 4c). At anthesis, well-watered Bahatans-87 had 50% and 65% more root biomass than Harper and Tincurrin, respectively *(p* < 0.001). In the ESD treatment, Bahatans-87 had 47% and 68% more root biomass than Harper and Tincurrin, respectively (*p* < 0.001; Figure 4d). ESD reduced total root biomass at anthesis by 33%, 29% and 38% in Bahatans-87, Harper and Tincurrin, respectively (*p* < 0.001; Figure 4d). 

At 30 DAS well-watered Bahatans-87 and Tincurrin had similar RLD, but Harper had 40% less (*p* < 0.001). In the ESD treatment at 30 DAS, all the three cultivars had similar low RLD (Figure 5a). In the first 30 DAS, ESD reduced RLD by 70% in Bahatans-87, 40% in Harper and 64% in Tincurrin (*p* < 0.001; Figure 5a). At anthesis, well-watered Bahatans-87 was 17% and 33% greater than in Harper and Tincurrin, respectively (*p* < 0.001; Figure 5b). ESD reduced RLD at anthesis by 33%, 26% and 46% in Bahatans-87, Harper and Tincurrin, respectively (*p* < 0.001; Figure 5b). 

At 30 DAS and under well-watered conditions, Bahatans-87 and Tincurrin had similar specific root lengths (SRL), while Harper had 17% more (*p* < 0.05; Figure 5c). In the ESD treatment at 30 DAS, the three cultivars had similar SRL (Figure 5c). In the first 30 DAS, ESD reduced SRL by 65% in Bahatans-87, 45% in Harper and 56% in Tincurrin (*p* < 0.05). At anthesis, well-watered Harper and Tincurrin had similar SRL, while Bahatans-87 had 40% less (*p* < 0.001; Figure 5d); similarly, in the ESD treatment, Harper and Tincurrin had similar SRL, while Bahatans-87 had 41% less (*p* < 0.001). The ESD treatment had no effect on SRL at anthesis in the three cultivars (Figure 5d). 

At 30 DAS, well-watered Tincurrin had 33% and 29% higher root to shoot ratios (R:S) than Bahatans-87 and Harper, respectively (*p* < 0.05); similarly, in the ESD treatment, Tincurrin had 8% and 13% higher R:S than Bahatans-87 and Harper, respectively (*p* < 0.05; Figure 6a). In the first 30 DAS, ESD increased the R: S in Bahatans-87, Harper and Tincurrin by 55%, 49% and 38%, respectively (*p* < 0.05; Figure 6b). At anthesis, well-watered Bahatans–87 had 46% and 36% higher R:S than Harper and Tincurrin, respectively (*p* < 0.001; Figure 6b); in the ESD treatment, Bahatans-87 had 43% and 17% higher R:S in Harper and Tincurrin, respectively (*p* < 0.001). ESD reduced R:S at anthesis by 32%, 28% and 11% in Bahatans-87, Harper and Tincurrin, respectively (*p* < 0.001; Figure 6b). 

### 3.5. Root Systems Distribution in the Soil Profile 

At anthesis, well-watered conditions Bahatans-87 had 54%, 54% and 93% more RLD than Tincurrin in the 0–0.2 m, 0.2–0.4 m and 0.8–1 m soil layers, respectively (*p* < 0.05; Figure 6c). At anthesis in the ESD treatment, Bahatans-87 and Tincurrin had similar RLD in the 0–0.2 m and 0.2–0.4 m soil layers, but Bahatans-87 had 35%, 93% and 98% more RLD than Tincurrin in the 0.4–0.6 m, 0.6–0.8 m and 0.8 –1.0 m soil layers, respectively (*p* < 0.01; Figure 6c). ESD reduced RLD at anthesis in Bahatans-87 by 28%, 51%, 48%, 39% and 54% in the 0–0.2 m, 0.2 –0.4 m, 0.4–0.6m and 0.8–1.0 m soil layers, respectively (*p* < 0.05). ESD had no effect on RLD at anthesis in Tincurrin in the 0–0.2 m and 0.2 –0.4 m soil layers, but reduced it by 39% and 95% in the 0.4–0.6 m, and 0.6–0.8 m soil layers, respectively (*p* < 0.01). At anthesis, well-watered Bahatans-87 had 54%, 72%, 58%, 44% and 96% more root biomass than Tincurrin in the 0–0.2 m, 0.2 –0.4 m, 0.4–0.6m and 0.8–1.0 m soil layers, respectively (*p* < 0.05); in the ESD treatment, the corresponding values were 46%, 43%, 50%, 98% and 99% (*p* < 0.05). ESD significantly reduced root biomass at anthesis in each 0.2 m soil layer (except 0.6–0.8 m) in Bahatans-87 (*p* < 0.05) and had on root biomass in the top 0–0.6 m in Tincurrin, but reduced by 96% in the 0.6–0.8 m soil layer (*p* < 0.01).

### 3.6. Grain Yield and Yield Components 

Well-watered Bahatans-87 and Tincurrin had similar grain yield, while Harper had 18.7% less (Table 3). A similar trend was observed in the ESD treatment. ESD reduced grain yield by 9.6%, 11.5% and 15.6% in Bahatans-87, Harper and Tincurrin, respectively (*p* < 0.001). Well-watered, Harper had 19% and 33% more spikes per plant than Bahatans-87 and Tincurrin, respectively (Table 3). In the ESD treatment, Bahatans-87 and Harper had similar spike numbers, while Tincurrin had 13% fewer (*p* < 0.01). Well-watered Bahatans-87 and Harper had similar grain numbers per spike under, while Tincurrin had 20% more (Table 3). In the ESD treatment, Tincurrin had 13% more grains per spike than Bahatans-87 and Harper. ESD had no effect on grain number per spike in Bahatans-87 and Harper, but reduced it by 16% in Tincurrin (*p* < 0.05). Well-watered Bahatans-87 had 27% and 7% 1000-grain weight than in Harper and Tincurrin, respectively (*p* < 0.01). In the ESD treatment, Bahatans-87 had 27% and 11% higher 1000-grain weight than Harper and Tincurrin, respectively (Table 3). ESD had no effect on 1000-grain weight in the three cultivars. Well-watered conditions, HI differed between the three cultivars, ranging from 0.35 to 0.52. In the ESD treatment, Bahatans-87 and Harper had similar HI but it was 24% higher in Tincurrin. ESD had no effect on HI in Harper, but reduced it by 8% and 12% in Bahatans-87 and Tincurrin, respectively (Table 3).

## 4. Discussion

### 4.1. Water Stress Characterisation of the Early Season Drought Treatment

The 25 mm of simulated rainfall applied to dry soil after sowing wet the top 0.12–0.15 m of the soil profile and allowed germination and emergence to occur. The subsequent restriction to water supply reduced the volumetric soil water content in the top 0.15 m layer of the soil profile. Although the plant water deficit was not measured as the volumetric soil water content decreased, the Ψ_leaf_—measured at 30 DAS, when the watering treatment had ended—had declined in the three cultivars with restricted water supply, relative to well-watered plants, indicating that water deficits had occurred. Ψ_leaf_ fell to –2.46 MPa in Bahatans-87 and –1.94 MPa in Harper and Tincurrin, both of which are lower than the Ψ_leaf_ at which stomatal conductance, transpiration rate and leaf photosynthetic rate in wheat are reduced (–1.4 MPa to –1.5 MPa) [28,29]. Indeed, at 30 DAS, g_s_ had declined in the three cultivars by about 90% (*p* < 0.001) and the rate of leaf photosynthesis by 65–83% (*p* < 0.001), indicating that the cultivars were under water stress. Since this physiological condition developed after plant emergence, we refer to it here as early season drought (ESD).

### 4.2. Early Season Drought Delayed Phenology in Wheat Cultivars with Different-Sized Root Systems 

In all three cultivars, ESD delayed time to booting, anthesis and physiological maturity. Tincurrin (small root system), reached anthesis 15 days earlier than Bahatans-87 (with large root system) independent of the water treatment, indicating faster phenological development in Tincurrin, which may explain its smaller root system and leaf biomass at anthesis compared to Harper and Bahatans-87. The delayed in phenology with ESD, when Ψ_leaf_ fell to about –2.5 MPa, could be due to a cessation of apical development and cell division [30]. Delaying the time to anthesis allows the root system to grow for a longer time [31], as the root system in wheat growth until anthesis [32,33,34]. However, delaying time to anthesis to compensate for the growth of the root system is not a useful strategy to increase yield in some environments, such as Mediterranean-type climates, where wheat crops often experience drought after flowering [35,36]; any delay to anthesis will reduce the opportunity escape the severe effects of terminal drought. 

### 4.3. Small Root System had Significant Reduction in Leaf Area and Biomass at Anthesis

ESD significantly reduced leaf area and shoot biomass at anthesis compare to well-watered plants, however no differences were observed in the ESD between the cultivars at 30 DAS in these parameters. At end of the ESD treatment, leaf area and shoot biomass recovered, more so in the cultivar with the large root system (Bahatans-87) than the cultivar with small root system (Tincurrin) because wheat cultivars with large root systems often have more days to booting [17] and anthesis [20] than those with small root systems, thus increasing the opportunity to accumulate more shoot biomass. However, leaf area and shoot biomass in Bahatans-87 did not recover to that of well-watered plants, indicating that ESD, despite delaying time to anthesis, restricted wheat growth. 

### 4.4. Early Season Drought Reduced the Root: Shoot Ratio at Anthesis

ESD reduced the root: shoot ratio at anthesis, regardless of the size of the root system. The reduction in root biomass at anthesis in the ESD treatment could be explained by the delay in the appearance of tillers [10], since root biomass and tiller number are close related [37]. In the vegetative stage during water deficit, wheat plants invest more in root biomass than shoot biomass [34,38]. However, during rewatering period the root growth rate in the ESD was lower compared to that in well water treatment [39], and the shoot growth rate between end of tillering and booting was the same or higher than the control [40]. These findings may explain why ESD reduced the root:shoot ratio at flowering. Root growth rated were not measured in this study, but the reduction in root biomass at anthesis could be due to slower rates of root growth during the recovery under well-water conditions [39], despite ESD delaying the time to anthesis, which increased the number of days for root growth. Differences between the cultivars in the time to anthesis after exposure to ESD may explain why root biomass at anthesis in Bahatans-87 (large root system) had less reduction in shoot biomass than Tincurrin. Despite the fact that Bahatans-87 and Harper have similar time to anthesis, the differences in root biomass may be due to the different root growth rate between the cultivars. The root:shoot ratio at anthesis in the cultivar with the large root system was reduced more than the cultivar with the smaller root system. This was mainly because Bahatans-87 and Harper took longer time to reach anthesis than Tincurrin and thus accumulated more shoot biomass. 

### 4.5. Early Season Drought Largely Reduced Grain Yield in a Cultivar with a Smaller Root System 

ESD reduced grain yield in all three cultivars, more so in Tincurrin, with the small root system. Tincurrin, had earlier tillering, booting and anthesis than cultivars with large root system [17,20]. Harper (medium root system) had less reduction in grain yield than Tincurrin. The percentage reduction on grain yield with ESD in Harper was higher than in Bahatans-87, indicating the longer phenology in Harper was not sufficient to compensate the grain yield reduction. Late heading and anthesis in wheat that has experienced ESD have been positive correlated with grain yield [41]. Selection for increased root system size in wheat showed that an increment of ~21% in root system size increased grain yield by 0.4 t ha^–1^ under rainfed conditions [14], indicating a positive correlation between root system size and grain yield in wheat growing under such conditions [15].

Breeding programs to improve wheat grain yield in dry environments and dry seasons have mainly focused on selecting for tolerance to terminal drought [42,43,44,45]. The selection of wheat cultivars with early growth and early anthesis has allowed wheat to escape the severe effects of terminal drought [46,47,48]. However, ESD—which has become more frequent in some dry environments such as the Mediterranean-type climate of Australia [2,49]—is delaying time to anthesis [7] and hence preventing wheat crops from escaping the severe effects of terminal drought has on grain yield.

## 5. Conclusions

The simulated 25 mm of rainfall that followed the planting into dry soil wet the top ~ 0.15 m of the soil layer, allowing germination and emergence to occur. The subsequent restricted water supply for 30 days (ESD) reduced Ψleaf, stomatal conductance, transpiration rate, leaf photosynthetic rate, root biomass and shoot biomass. ESD delayed phenology, particularly time to anthesis, with time to anthesis delayed more in Bahatans-87 (large root system) than Tincurrin (small root system) allowing more time to recover leaf area and shoot biomass. It is likely that the faster phenological development in the cultivar with the small root system exposed it to more to ESD because at the end of 30 DAS Tincurrin was close to double ring stage. An important question arising from this study is to what extent the delay in time to anthesis in wheat experiencing ESD reduce the severe effects of terminal drought and its effect on grain yield.

## Figures and Tables

**Figure 1 plants-08-00305-f001:**
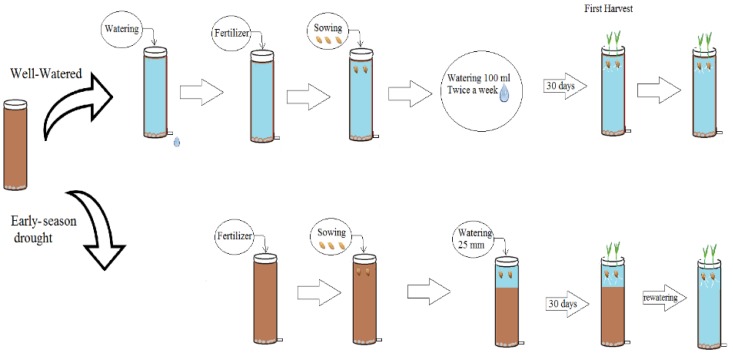
Diagram of the watering treatments in three wheat cultivars with different root systems, grown under well-watered conditions and exposed to early season drought (ESD), which was induced by simulating a 25 mm rainfall after sowing in dry soil and no watering for 30 days.

**Figure 2 plants-08-00305-f002:**
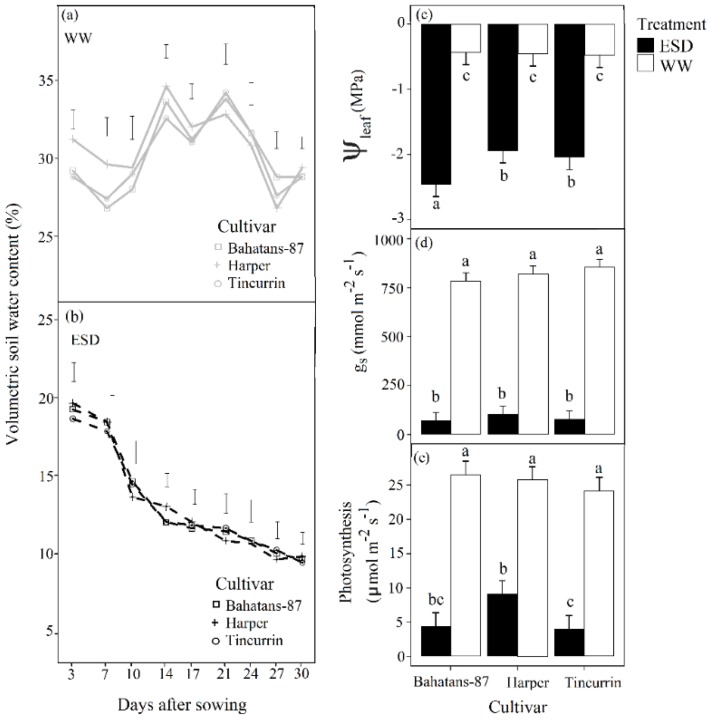
Volumetric soil water content (%) in (**a**) well-watered treatment and (**b**) early season drought (ESD) measured in the top 0.15 m of the soil profile during the watering treatments; (**c**) leaf water potential (Ψleaf), (**d**) stomatal conductance (gs), and (**e**) leaf photosynthesis rate measured at the end of the early-season drought treatment (30 DAS) in three cultivars with different root systems, grown under well-watered conditions (WW) and exposed to ESD. ESD was induced by simulating a 25 mm rainfall after sowing in dry soil and no watering for 30 days. Mean values (n = 30) followed by different letters are significantly different *p* < 0.05. Vertical error bars represent s.e. of the mean (n = 5).

**Figure 3 plants-08-00305-f003:**
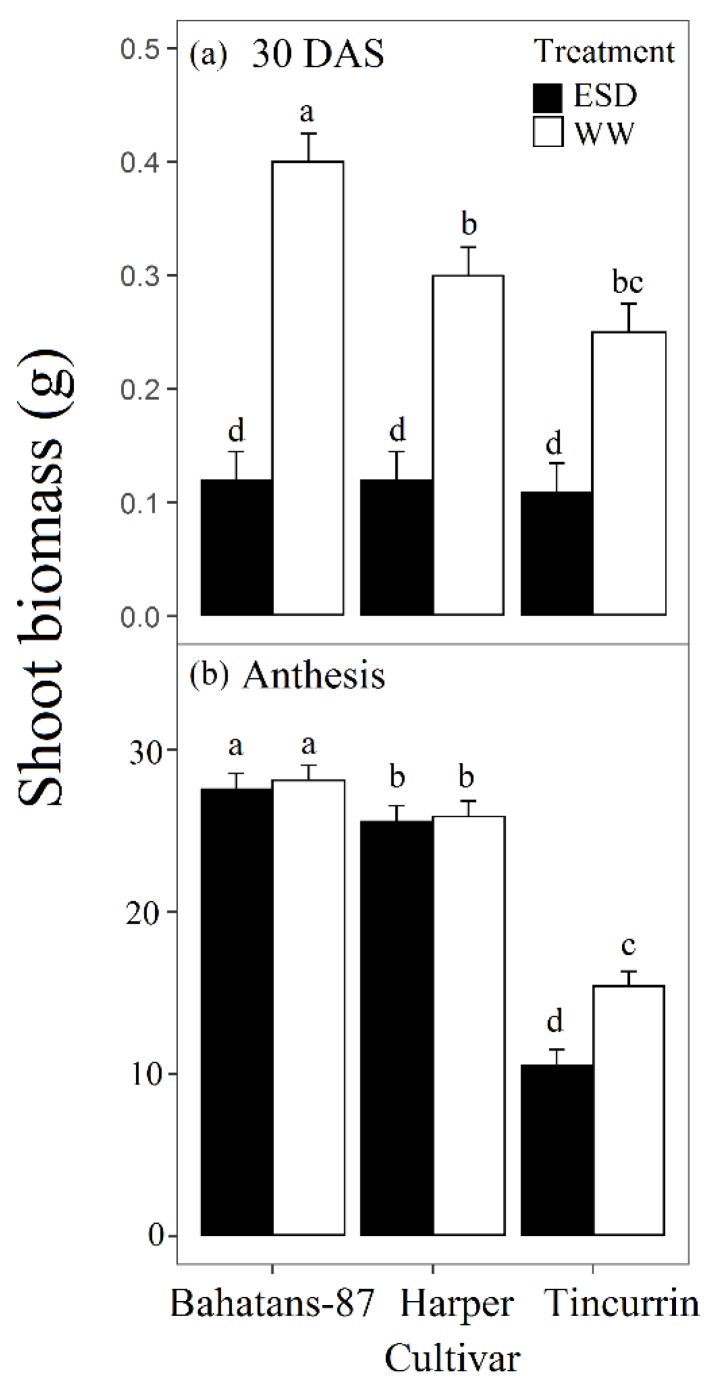
Shoot biomass at (**a**) the end of the drought treatment (30 DAS) and (**b**) anthesis, in three wheat cultivars with different root systems, grown under well-watered conditions (WW) and exposed to early season drought (ESD). ESD was induced by simulating a 25 mm rainfall after sowing in dry soil and no watering for 30 days. Means followed by different letters are significantly different *p* < 0.05. Vertical error bars represent s.e. of the mean (n = 5).

**Figure 4 plants-08-00305-f004:**
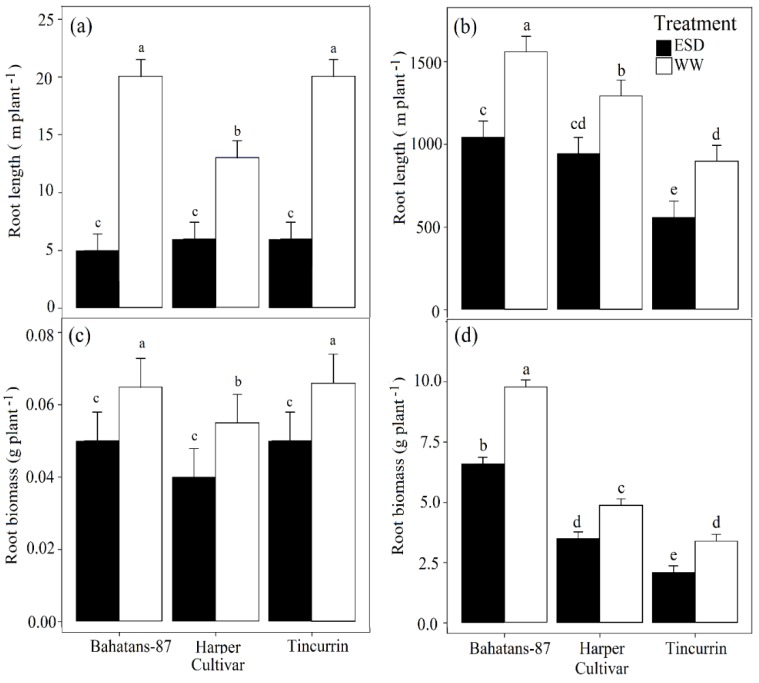
Total root length at (**a**) end of the drought treatment (30 DAS) and (**b**) anthesis; total root biomass at (**c**) end of the drought treatment and (**d**) at anthesis in three wheat cultivars with different root systems, grown under well-watered conditions (WW) and exposed early season drought (ESD). ESD was induced by simulating a 25 mm rainfall after sowing in dry soil and no watering for 30 days. Means followed by different letters are significantly different *p* < 0.05. Vertical error bars represent s.e. of the mean (n = 5).

**Figure 5 plants-08-00305-f005:**
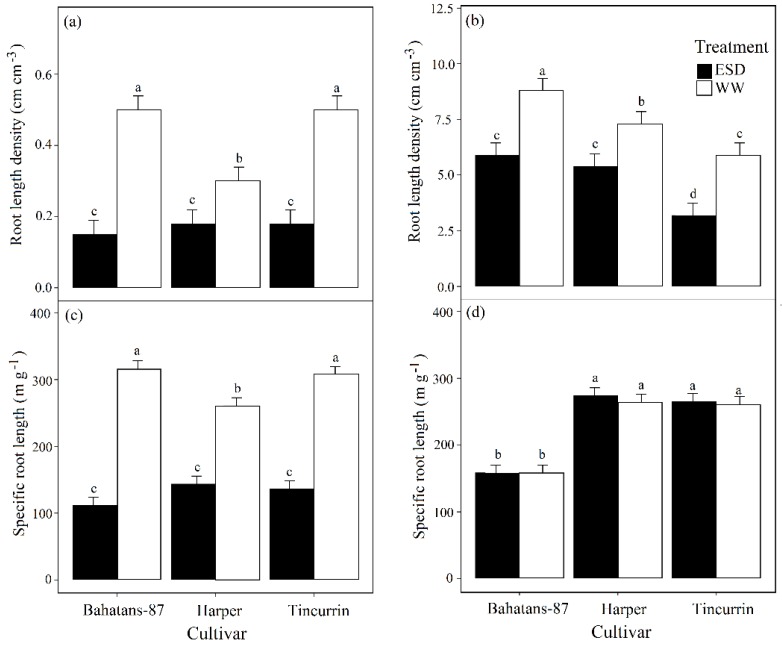
Root length density at (**a**) end of the drought treatment (30 DAS) and (**b**) anthesis, and specific root length (**c**) at the end of the drought treatment and (**d**) at anthesis in three wheat cultivars with different root systems, grown under well-watered conditions (WW) and exposed to early season drought (ESD). ESD was induced by simulating a 25 mm rainfall after sowing in dry soil and no watering for 30 days. Means followed by different letters are significantly different *p* < 0.05. Vertical error bars represent s.e. of the mean (n = 5).

**Figure 6 plants-08-00305-f006:**
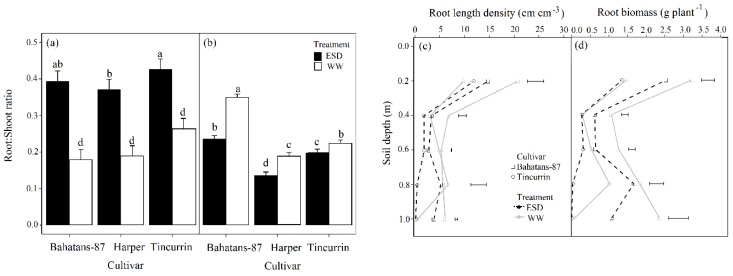
Root: Shoot ratio at (**a**) the end of the drought treatment (30 DAS) and (**b**) anthesis; (**c**) the distribution of root length density, and (**d**) root biomass down the soil profile in wheat cultivars with different root systems, grown under well-watered conditions (WW) and exposed to early season drought (ESD). ESD was induced by simulating a 25 mm rainfall after sowing in dry soil and no watering for 30 days. Means followed by different letters across cultivars and treatments in (**a**) and (**b**) are significantly different at *p* < 0.05. Vertical error bars in (**a**) and (**b**) represent s.e. of the mean (n = 5). Horizontal bars in (**c**) and (**d**) represent LSD at *p* < 0.05 for comparison between cultivars at each depth in the soil profile.

**Table 1 plants-08-00305-t001:** Number of days to booting (Z49), anthesis (Z61), physiological maturity (Z91) and duration of the grain filling in three wheat cultivars with different root systems, grown under well-watered conditions (WW) and exposed to early season drought (ESD). ESD was induced by simulating a 25 mm rainfall after sowing in dry soil and no watering for 30 days.

Cultivar	Booting		Anthesis		Physiological Maturity		Duration of the Grain Filling	
	(DAS)		(DAS)		(DAS)		Days	
	WW	ESD	WW	ESD	WW	ESD	WW	ESD
Bahatans-87	93c	100b	102c	109b	139c	150b	37c	41c
Harper	100b	110a	107b	120a	170a	174a	63a	54b
Tincurrin	74e	81d	87e	94d	137c	147b	50b	53b
LSD (C) LSD (T) LSD (CxT)	2.3 1.9 3.3**		2.9 2.4 4.1**		6.1*** 5** NS		4.1 3.4 5.8***	

Means followed by different letters are different according to least significant difference (LSD) test. **, *** significant at *p* < 0.01 and *p* < 0.001, respectively. Cultivar (C) and Treatment (T).

**Table 2 plants-08-00305-t002:** Leaf area, specific leaf area (SLA) and tiller number measured when the early season drought treatment was ended at 30 DAS, and 50% anthesis (Z61) in in three wheat cultivars with different root systems, grown under well-watered conditions (WW) and exposed to early season drought (ESD). ESD was induced by simulating a 25 mm rainfall after sowing in dry soil and no watering for 30 days.

Cultivar	Leaf Area (cm^2^plant^−1^)				Specific Leaf Area (cm^2^ g ^−1^)				Tiller Number (plant^−1^)			
	30 DAS		Anthesis		30 DAS		Anthesis		30 DAS		Anthesis	
	WW	ESD	WW	ESD	WW	ESD	WW	ESD	WW	ESD	WW	ESD
Bahatans-87	101a	20c	1946b	1648c	270b	159d	229c	215bc	3a	0d	17a	14b
Harper	75b	24c	2604a	2444a	272b	212c	245b	225b	2b	0d	18a	16b
Tincurrin	77b	24c	1548c	1194d	307a	219c	305a	287a	1c	0d	8c	7c
LSD (C)	8.4		179.7		17.6		13.2		0.2		1.2***	
LSD (T)	6.9		146.7		14.4		10.7		0.2		1.0**	
LSD (CxT)	11.9**		254.1*		24.9*		18.6***		0.3***		NS	

Means followed by different letters are different according to the least significant difference (LSD) test. *, **, *** significant at *p* < 0.05, *p* < 0.01 and *p* < 0.001, respectively. NS, no significant *p* > 0.05. Cultivar (C) and Treatment (T).

**Table 3 plants-08-00305-t003:** Grain yield, number of spikes per plant, number of grain per spike, 1000-grain weight and harvest index (HI) in three wheat cultivars with different root systems, grown under well-watered conditions (WW) and exposed to early season drought (ESD). ESD was induced by simulating a 25 mm rainfall after sowing in dry soil and no watering for 30 days.

Cultivar	Grain Yield		Spikes		Grains		1000 Grain Weight		Harvest Index	
	(g plant^−1^)		(plant^−1^)		(spike^−1^)		(g)			
	WW	ESD	WW	ESD	WW	ESD	WW	ESD	WW	ESD
Bahatans-87	31a	28b	17b	16b	42c	40c	44a	45a	0.38c	0.35d
Harper	26c	23d	21a	16b	44bc	40c	32c	33c	0.35d	0.34d
Tincurrin	32a	27bc	14c	14c	55a	46b	41b	40b	0.52a	0.46b
LSD (C)	2.4***		1.3		3.7		1.6**		0.03***	
LSD (T)	1.9***		1.1		3.0		NS		0.03*	
LSD (CxT)	NS		1.8**		5.2*		NS		NS	

Means followed by different letters are different according to the least significant difference (LSD) test. *, **, *** significant at *p* < 0.05, *p* < 0.01 and *p* < 0.001, respectively. NS, no significant *p* > 0.05. Cultivar (C) and Treatment (T).

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
