# Peer review of "Early Season Drought Largely Reduces Grain Yield in Wheat Cultivars with Smaller Root Systems"

_plants, 2019, doi:10.3390/plants8090305_

Round 1

Reviewer 1 Report

Climate change is altering the pattern of rainfall and as a consequence the farmers’ habits. This paper address a particular problem related to a drought period in the cultivation of some wheat varieties in Australia.
In this manuscript, Figueroa-Bustos and co-workers thoroughly describe the effects of early-season drought in many physiological and phenotypic traits in three different wheat cultivars. Although the manuscript is largely descriptive, in general, the methodology, the way the authors show the results and the conclusions are clear. There are, however, small details that in my opinion should be addressed before publication.
Line 313-317: About the structure of the manuscript. In my opinion, the information shown in the first part of the methods section should be move to the end of the introduction or the beginning of the results section.
Lines 335-354. Similarly to my previous comment, this part is essential to understand the whole manuscript. I suggest to move this part to the beginning of the manuscript and transform Figure 6 into Figure 1.
Line 397: Again about the structure. The conclusions section should not be after the methods section. The latter should be the first or the last in a manuscript.
Line 11: here is the first time the authors define “early season drought”. From here, these three words are said 6 times in the abstract, 5 times in the introduction and more than 60 times in the results part. I suggest to mention and fully explain the concept at the beginning and from there use the acronym ESD (as in Figures)to facilitate the reading.
Line 39: there is an incorrect full stop here.
Lines 64, 96, 233, 158 and 407. The mistake of not defining a concept the first time the authors mention it in the manuscript is recurrent all along with it.
Figure Legends: I do not see necessary to define how the ESD treatment was performed in all the figure legends. I suggest to keep it in the first one and reference it to it in the rest of them.
The panel (a) in Figure 1 is not clear to me. There is a lot of space and all the data points are very close. I suggest to split the information into two different graphs or to represent it differently.
Figures: I wonder why all the bars charts lack error bars. I assume that authors have performed different measurement and the dispersion of the data is also information beyond the statistical analysis.
Line 87: There are two full stops in a row.
Table 1 and Table 2: I wonder if the data shown in these tables would be more visual and easy to understand in Figures instead.
About Figure 2: the Y-axis name should be completed. One is at 30 das and the other at anthesis.
Lines 61, 67, 183, 251, 274, 295, 297, 326, 360, 361: There are double spaces. Please revise all the manuscript.
Line 322: there should be a space between mixture and (75:25).
Line 135: “7 m” instead of “7m”.
Bibliography:
The authors have mixed abbreviations and full journal names. Please be consistent and follow the journal guidelines.

Reviewer 2 Report

Manuscript is written well and clearly stated the objective and rationale for the study. However, there are some general comments which could improve the article. 1.It is not clear why the paper has results section right after introduction instead of materials and methods which is a bit odd and confusing. 2.There are three cultivars tested and it appears that in some sections of the discussion (section 3.4, 3.5) cultivar Harper was not discussed enough 3.Appreciated for thoroughly describing the materials and methods. It was noted that fertilizer was incorporated in the soil for both treatments (WW, ESD) however while imposing the drought cycle how did you make sure they were not nutrient stress? 4.Entire research was done in a controlled environment and the data was from a single run or year. It would be good to repeat the study and evaluate the results; will add more power (statistical analysis).Data from multiple years or runs can be useful while defending the controlled environment research because often there will be a question on translatability of the results to field environment?

Reviewer 3 Report

This is a very carefully done study which is marred by a poor presentation. The introduction is OK and the methods are well-described, but the results are almost unreadable. The results should not just repeat what is in the tables and graphs, but should try to provide an overall picture of the important findings.  The use of the LSD is confusing and, I think, erroneous. t would be better to report the ANOVA tables for each variable.  For treatment versus control, the LSD is not necessary as the ANOVA provides the same information. For the three populations and the interaction means, it would be better to use Tukey's Honestly Significant Difference (HSD) to distinguish between means. If there is a significant interaction, then it is inappropriate to ascribe much importance to significant differences between the main effects.  

  The discussion could be improved by relating the findings to the extensive literature on investment on root versus shoot tissue. There are many grammatical mistakes in the discussion (notably lines 261-262, 264) as well as elsewhere. The whole ms. should be carefully reviewed by a native speaker of English. 

Round 2

Reviewer 3 Report

The results section remains hard to follow. And the use of the LSD is inappropriate given the large number of comparisons. Nevertheless, the findings are clear enough that statistical mistakes do not obscure the overall message. Some minor errors in English remained in the discussion, so I fixed them in the attached PDF file. 
